# Development and Validation of a Real-Time Service Model for Noise Removal and Arrhythmia Classification Using Electrocardiogram Signals

**DOI:** 10.3390/s24165222

**Published:** 2024-08-12

**Authors:** Yeonjae Park, You Hyun Park, Hoyeon Jeong, Kise Kim, Ji Ye Jung, Jin-Bae Kim, Dae Ryong Kang

**Affiliations:** 1Department of Medical Informatics and Biostatistics, Graduate School, Yonsei University, Seoul 03722, Republic of Korea; aceail@yonsei.ac.kr (Y.P.); dbgus6175@yonsei.ac.kr (Y.H.P.); hoyeonjeong@yonsei.ac.kr (H.J.); 2National Health BigData Clinical Research Institute, Yonsei University Wonju Industry-Academic Cooperation Foundation, Wonju 26426, Republic of Korea; 3School of Health and Environmental Science, Korea University, Seoul 02841, Republic of Korea; 2019250246@korea.ac.kr; 4Division of Pulmonary and Critical Care Medicine, Department of Internal Medicine, Severance Hospital, Yonsei University College of Medicine, Seoul 03722, Republic of Korea; stopyes@yuhs.ac; 5Division of Cardiology, Department of Internal Medicine, Kyung Hee University Hospital, School of Medicine, Kyung Hee University, Seoul 02447, Republic of Korea; jinbbai@khu.ac.kr; 6Department of Precision Medicine and Biostatistics, Yonsei University Wonju College of Medicine, Wonju 26426, Republic of Korea

**Keywords:** electrocardiogram denoising, generative adversarial network, arrhythmia classification, wearable device

## Abstract

Arrhythmias range from mild nuisances to potentially fatal conditions, detectable through electrocardiograms (ECGs). With advancements in wearable technology, ECGs can now be monitored on-the-go, although these devices often capture noisy data, complicating accurate arrhythmia detection. This study aims to create a new deep learning model that utilizes generative adversarial networks (GANs) for effective noise removal and ResNet for precise arrhythmia classification from wearable ECG data. We developed a deep learning model that cleans ECG measurements from wearable devices and detects arrhythmias using refined data. We pretrained our model using the MIT-BIH Arrhythmia and Noise databases. Least squares GANs were used for noise reduction, maintaining the integrity of the original ECG signal, while a residual network classified the type of arrhythmia. After initial training, we applied transfer learning with actual ECG data. Our noise removal model significantly enhanced data clarity, achieving over 30 dB in a signal-to-noise ratio. The arrhythmia detection model was highly accurate, with an F1-score of 99.10% for noise-free data. The developed model is capable of real-time, accurate arrhythmia detection using wearable ECG devices, allowing for immediate patient notification and facilitating timely medical response.

## 1. Introduction

The different organs of the human body send signals indicative of their functional status. While some symptoms, such as cough or fever, visibly manifest and quickly alert us to health issues, others are not as apparent. Certain diseases, such as stroke and cardiovascular disease (CVD), which are responsible for approximately 18.6 million deaths globally each year, may progress without any external symptoms [1]. This underscores the critical need for ongoing research into cardiovascular health, with an emphasis on early detection and continuous monitoring.

Early intervention for CVD is vital for improving patient outcome and reducing mortality rates. The electrocardiogram (ECG), which is a noninvasive tool, plays a crucial role in this endeavor by monitoring the electrical activity of the heart. It provides invaluable data for diagnosing, classifying, and tracking heart diseases, aiding healthcare professionals in devising effective treatment plans [2].

Conventionally, a 24 h Holter monitor is employed to record heart activity over the period of a day. However, irregularities that occur during routine activities may be missed due to its limited monitoring window, and its usage can be cumbersome in daily life. Fortunately, advancements in technology have enabled continuous ECG monitoring through wearable devices [3].

However, ECG signals from wearable devices are prone to contain various types of noise, such as exercise-induced disturbances and electrical interference, as well as artifacts from muscle movements or poor electrode contact. These interferences can significantly hinder an accurate ECG analysis [4].

Conventional noise removal techniques have primarily involved approaches such as linear filtering and wavelet transformations [5,6,7]. However, these methods often fail to meet real-time processing requirements and cannot adequately consider the limited computational power of wearable devices. In particular, conventional methods are associated with significant limitations in effectively separating and eliminating complex noise patterns. The various types of noise that can be found in the data collected by wearable devices, such as motion-induced noise or electrical interference, can be highly diverse and difficult to predict [8]. These issues cannot be sufficiently addressed using conventional filtering methodologies, making it necessary to develop advanced noise removal technologies in the field of electrocardiogram signal processing. With the advent of deep learning, innovative approaches, such as autoencoders and generative adversarial networks (GANs), have become popular owing to their enhanced efficacy for noise reduction [9,10]. These advanced models offer significant improvements over conventional methods, particularly in their ability to more effectively adapt to and process complex and unpredictable types of noise.

Noise removal from ECGs and the classification of arrhythmias using clean ECGs have been studied extensively, and several studies have compared the performances of different models rather than fitting models to actual measurement data [11,12,13]. Additionally, actually applied studies exist, but there are a limited number of studies that eliminated noise in real time and classified arrhythmias using artificial intelligence (AI) [14,15].

This study introduces a novel model that employs generative adversarial networks (GANs) for robust noise removal and utilizes ResNet for accurate arrhythmia classification from ECG signals. By employing two databases, namely MIT-BIH Arrhythmia and MIT-BIH Noise Stress Test databases, we not only investigated the potential of our model for improved noise filtration and arrhythmia detection but also validated its practical application to real-world wearable device data. In contrast to previous research, this study directly used the measured data for external validation. The noise removal performance was verified through simple evaluation metrics, as well as via comparative analysis of the signals with noise, original signals, and noise-removed signals for arrhythmia classification. Our findings demonstrate the potential of technological advancements in real-time ECG monitoring and disease diagnosis, enabling transformative improvements in cardiac health management.

## 2. Methodology

### 2.1. Data Acquisition

#### 2.1.1. Public Data

We used the MIT-BIH Arrhythmia and Noise Stress Test databases provided by PhysioNet. The MIT-BIH Arrhythmia database was compiled between 1975 and 1979 using the Holter method. As listed in Table 1, it comprises 47 records, corresponding to 360/s measurements with 11-bit resolution in the 10 mV range for 30 min [16]. The MIT-BIH Noise Stress Test database contains three types of noise: baseline wander (BW), motion artifact (MA), and electrode motion (EM) artifact [17].

#### 2.1.2. Measurement Data Collected in a Practical Environment

In this study, we utilized the Hicardi^®^ (MEZOO Co., Ltd., Wonju-si, Gangwon-do, Republic of Korea), a wearable ECG monitoring patch weighing 8 g and measuring 42 mm × 30 mm × 7 mm. This device is certified by the Korea Food and Drug Safety Agency and captures ECG signals at a sampling frequency of 250 Hz, with a resolution of 14 bits. It can monitor and record single-lead ECG, respiration, skin surface temperature, and physical activity. Data from the wearable patch were transmitted to a mobile gateway via Bluetooth Low Energy, implemented as a portable smartphone application. All collected data were then forwarded to a cloud-based monitoring server through the mobile gateway. After obtaining informed consent from all participants, the wearable patch was affixed to the left sternal border, and continuous recording of ECG signals and other data ensued. These data were subsequently reviewed by cardiologists using the cloud-based monitoring system. The data were measured at 250 Hz but were up-sampled to 360 Hz using the Fourier method to match the frequency of the data in the MIT-BIH database. This adjustment ensures the temporal consistency of the input data and secures a uniform input shape. Using this method, we measured the ECG for 3 d in 35 participants, after obtaining informed consent. Table 1 presents the characteristics of the subjects in terms of the open and measured data.

### 2.2. Noise Removal from ECG Signals

In this study, the generator of the GAN model was constructed based on U-Net and residual blocks [18,19]. U-Net is an end-to-end fully convolutional network-based model and has been used for image segmentation in the biomedical field. It can solve the trade-off issue between localization and context through its skip architecture concept and by combining feature maps of shallow and deep layers. However, it is difficult to construct a deep network using U-Net alone. To overcome this limitation, we efficiently built deep networks using residual blocks instead of convolutional layers. The discriminator of the GAN comprises convolutional layers, batch normalization, and leaky ReLU for quick discrimination (Figure 1).

Problems such as mode collapse or a vanishing gradient occur during GAN training [20,21]. To address these problems, the least-squares GAN (LSGAN) framework has been proposed [22]. In this approach, the original binary cross-entropy loss function is replaced with the least-squares function. The loss function for the LSGAN model is mathematically represented in Equations (1) and (2).
(1)$$\underset{D}{min}V_{LSGAN}\left(D\right)=\frac{1}{2}E_{x\sim p_{data}\left(x\right)}\left[{\left(D\left(x\right)-1\right)}^{2}\right]+\frac{1}{2}E_{z\sim p_{z}\left(z\right)}\left[{\left(D\left(z\right)\right)}^{2}\right]$$
(2)$$\underset{G}{min}V_{LSGAN}\left(G\right)=\frac{1}{2}E_{z\sim p_{z}\left(z\right)}\left[{\left(D\left(G\left(z\right)\right)-1\right)}^{2}\right]$$

We trained the model based on the least squares of the LSGAN loss function. Learning was performed based on the loss function of the LSGAN. In this regard, we used Equation (3) to make the generator more accurate. The Euclidean distance function is used to determine the difference between real data and data produced by the generator as the loss function. The smaller the Euclidean difference, the more similar the generated data to the real data. However, this is the average of the difference in the distance, which results in a weakness for the outliers. This limitation is solved using Equation (4), which is a method for obtaining the maximum value rather than the average of the difference in the distance between the generated and real data. Unlike $L_{dist}$, the difference between the two data distances can be compensated for outliers with absolute values. To ensure that this occurs, the value for the outlier is corrected using the maximum of the distance between the two sets of data. By adding this information to the generator loss in the LSGAN, we can derive the final expression as Equation (5). In Equations (3)–(5), n is the number of signal samples, $s_{k}\;$represents the clean signal, ${\hat{s}}_{k}$ is the denoised signal, a = 0.7, and b = 0.3 [23].
(3)$$L_{dist}=\sqrt{\sum_{k=1}^{n}{\left({\hat{s}}_{k}-s_{k}\right)}^{2}}$$
(4)$$L_{\mathrm{dist}-\max}=\max\left(\left|{\hat{s}}_{1}-s_{1}\right|,\left|{\hat{s}}_{2}-s_{2}\right|,\ldots ,\left|{\hat{s}}_{n}-s_{n}\right|\right)$$
(5)$$L_{G}=\frac{1}{2}E_{z\sim p_{z}\left(z\right)}\left[{\left(D\left(G\left(z\right)\right)-1\right)}^{2}\right]+\alpha \cdot L_{dist}+\beta \cdot L_{\mathrm{dist}-\max}$$

### 2.3. Arrhythmia Classification Using ECG Signals

ResNet exhibited high performance in several previous classification models [24]. Numerous studies have proposed approaches that offer high performance with regards to arrhythmia classification using ECG signals. The ResNet residual block comprises several essential components. First, each residual block includes an identity shortcut that bypasses one or more convolutional layers. This shortcut ensures that information is preserved as it passes through the layers by directly adding the input of the block to its output, which helps mitigate the vanishing gradient problem during training. Second, each residual block typically contains two convolutional layers that are crucial for feature extraction and transformation. The first convolutional layer processes the input data, followed by batch normalization and a ReLU activation function. Subsequently, the second convolutional layer further processes the data, followed by another batch normalization and ReLU activation. The output of this second layer is then added to the original input to produce the block’s output. This architecture aids in alleviating the vanishing gradient problem that can occur as the network deepens, thus enabling the effective training of deeper networks. Therefore, we used ResNet in this study.

### 2.4. Service Utilization of Developed Model

The wearable device (Mezoo HiCardi SmartPatch; Mezoo Co. Ltd., RM.808 200, Gieopdosi-ro, Jijeong-myeon, Wonju-si, Gangwon-do, Republic of Korea) collects real-time ECG data from the body, which is then transmitted via Bluetooth to a smartphone app. This app serves as a user interface and forwards the data to the web server via API communication (Figure 2A). The web server securely transmits the data to the broker server using the HTTPS protocol. The broker server, functioning as a streaming broker and a REST API server (Figure 2B), initiates data analysis by streaming the data to the GPU server. Here, the data undergo parsing, slicing, normalization, noise removal, and arrhythmia classification to predict the final results (Figure 2C). The processed results are then streamed back through the broker server and displayed on the smartphone, enabling real-time ECG monitoring. The system employs Airflow (v.2.2.5) for data flow management, Kafka (v.2.12–2.8.1) as the broker server, and TensorFlow (v.2.4.1) as the deep learning framework. The overall system architecture is illustrated in Figure 3.

## 3. Experimental Configuration

### 3.1. Experimental Preparation

This study entailed several integrated steps: data sourcing and preprocessing, noise removal, bit segmentation, arrhythmia classification, and performance evaluation for both noise removal and ECG arrhythmia classification. Initially, the original ECG signals, sourced from public databases (MIT-BIH Arrhythmia and MIT-BIH Noise Stress Test databases from PhysioNet) and wearable devices, undergo preprocessing. During this phase, a time-series data window sliding technique is utilized, signals are normalized using the min–max method, and the noise removal model is trained and evaluated using specific indices. Subsequently, for the arrhythmia classification model, the preprocessed signals are segmented into bits using the time-series data window sliding technique. These bits are then trained using a deep learning model based on ResNet to classify arrhythmias, with verification based on evaluation indices. To assess the difference in arrhythmia classification performance between the original and noise-removed signals, a proposed GAN model is employed to eliminate noise. Thereafter, the noise-removed signal is processed through the same ECG beat segmentation step and trained using the existing arrhythmia classification model, followed by model verification based on evaluation indices. For practical application, noise is removed from directly measured data using the same method, and the existing arrhythmia classification model is adapted to finalize the model. The completed model is evaluated based on various performance indicators to assess its real-world applicability. This process is depicted in Figure 4, which visually outlines the flow and methodology of each step.

#### 3.1.1. Experimental Environment

All experiments were performed on a workstation with an AMD Ryzen thread ripper PRO 3975WX processor (AMD, Santa Clara, CA, USA) with 32 cores, hyper-threaded to 64 virtual cores, a base clock frequency of 3.5 GHz CPU, and an NVIDIA RTX A6000 GPU (NVIDIA, Santa Clara, CA, USA). The training was conducted under PyCharm IDE with Python (v.3.8.13), TensorFlow (v.2.8.0), and Keras (v.2.8.0).

#### 3.1.2. Data Preprocessing

In this study, the MIT-BIH Arrhythmia and MIT-BHI Noise Stress Test databases were used. The MIT-BIH Arrhythmia database contains recordings of 47 people. Most channels in the record provide modified limb lead II (MLII). However, the records without MLII (i.e., record numbers 102 and 104) were excluded from this study. The MIT-BIH Noise Stress Test database uses BW, MA, and EM noise. For the denoising model, raw data were obtained from the same 10 records, numbered 103, 105, 111, 116, 112, 205, 213, 219, 223, and 230, in the MIT-BIH Arrhythmia database for comparison with the data from the proposed method. It is necessary to determine the amount of noise when synthesizing the three types of noise in the MIT-BIH Arrhythmia database. The magnitude of the noise to be synthesized was determined by the signal-to-noise ratio (SNR) and was applied by multiplying the noise and the weight. The level of noise interference was applied at two noise levels, 0 dB and 5 dB, determined by the signal-to-noise ratio (SNR). Here, Equation (6) expresses the weighted noise synthesis, where *t*, *N*(*t*), and *S*(*t*) indicate time, noise signal, and clean signal, respectively. The noise weight is expressed as using Equation (7).
(6)$$N\left(t\right)=S\left(t\right)+\alpha \cdot n\left(t\right)$$
(7)$$\alpha =\sqrt{\frac{\sum_{t=1}^{T}S^{2}\left(t\right)}{10^{\frac{\mathrm{SNR}}{10}}\sum_{t=1}^{T}n^{2}\left(t\right)}}$$

Among the labels provided in the MIT-BIH Arrhythmia database for the classification model, only the normal beat (N), left branch block beat (L), right branch block beat (R), premature ventricular contraction (V), atrial premature beat (A), and ventricular fusion beats and normal beats (F) were used. The block beats of the left and right branches were treated as one category of the interventricular block (B). The data were segmented by time and not by the R-peak. For denoising, the original data and each noise segment are shifted to 180 Hz (0.50 s) and then cut at 1024 Hz (2.84 s) to synthesize the noise. Additionally, the segmented original data is divided into two intervals, 0–512 Hz and 513–1024 Hz, each at 512 Hz (1.42 s), for arrhythmia classification. This method was configured in such a way as to eliminate noise in real-time and to classify arrhythmia. However, in this case, two or more types of ECG beats may simultaneously exist at 512 Hz (1.42 s). In this case, the type of ECG beat that occurred first was selected. This configuration provides 2.73 times more data than the amount procured using existing labels, without rolling. We split the data in the ratio of 8:2 using the proposed method for training and testing (Table 2). The real-life ECG measurement data excluded the lead-off portion of the measurement due to the subject. Ventricular fusion beats and normal beats did not occur during reading.

### 3.2. Evaluation Metrics

We used the following metrics to evaluate the model performance: percent root-mean-squared difference (PRD (%), (8)), SNR (9), root-mean-squared error (RMSE, (10)), positive predictive value (PPV, also known as precision, (9)), sensitivity (as known as Recall, (11)), and F1-score (13). Herein, *n* is the number of samples for the evaluation, *x*(*n*) represents the raw signal, $\hat{x}\left(n\right)$ $\hat{x}$(*n*) represents the denoised signal, TP is the true negative, FP is the false positive, and FN is the false negative. Each of these metrics allows for the evaluation of various quantitative aspects of the denoised signal against the clean ECG signal. Thus, the accuracy of arrhythmia classification can be confirmed. Higher Precision, Recall, and F1-score values indicate a better performance.
(8)$$\mathrm{PRD}(\%)=\sqrt{\frac{\sum_{n=1}^{N}{\left(x\left(n\right)-\hat{x}\left(n\right)\right)}^{2}}{\sum_{n=1}^{N}x^{2}\left(n\right)}}\times 100$$
(9)$$\mathrm{SNR}=10\log_{10}\frac{\sum_{n=1}^{N}{\left(x\left(n\right)\right)}^{2}}{\sum_{n=1}^{N}{\left(\hat{x}\left(n\right)-x\left(n\right)\right)}^{2}}$$
(10)$$\mathrm{RMSE}=\sqrt{\frac{1}{N}\sum_{n=1}^{N}{\left(\hat{x}\left(n\right)-x\left(n\right)\right)}^{2}}$$
(11)$$\mathrm{Precision}=\frac{\sum \mathrm{TP}}{\sum \left(\mathrm{TP}+\mathrm{FP}\right)}$$
(12)$$\mathrm{Recall}=\frac{\sum \mathrm{TP}}{\sum \left(\mathrm{TP}+\mathrm{FN}\right)}$$
(13)$$F1\;\mathrm{Score}=2\times \frac{\mathrm{Recall}\times \mathrm{Precision}}{\mathrm{Recall}+\mathrm{Precision}}$$

## 4. Results

### 4.1. Denoising of Public ECG Data

To evaluate the performance of the denoising model, the BW, EM, and MA noise types were studied at noise levels of 0 and 5 dB. We compared the proposed model using only the LSGAN loss function against the LSGAN with U-Net. The proposed model exhibited an SNR of more than 30 dB for all types of noise. By contrast, when only U-Net was used, the SNR did not exceed 20 dB for any of the noise types. When using only the LSGAN loss function, the SNR was close to 20 dB (Table 3). The results for the proposed model are shown in Figure 5. Additional details are provided in Appendix A. Figure 5 synthesizes Appendix A to visually compare the performance of the proposed model under different noise types and levels.

We evaluated the proposed noise removal model for signals contaminated with mixed noise, in addition to the BW, EM, and MA noise types. The different types of noise signals used in the synthesis were 0.3∙BW + 0.7∙MA, 0.3∙EM + 0.7∙MA, 0.5∙BW + 0.5∙EM, and 0.25∙BW + 0.25∙EM + 0.5∙MA. During the synthesis process, we assigned higher importance to the MA noise by assigning it a higher weight, as we believed it to be the most significant type of noise. We confirmed that all the synthesized noise exhibited an SNR of over 30 dB, based on the evaluation results (Table 4). The results for the proposed model are shown in Figure 6. Additional details are provided in Appendix A. Figure 6 synthesizes Appendix A to visually compare the performance of the proposed model under different noise types and levels.

The results show that the noise is not properly removed when only U-Net is used. Moreover, noise can be removed at a higher level of performance if a loss function with distance information is added to the generator loss function.

### 4.2. Arrhythmia Classification Using Public ECG Data

To evaluate the performance of the arrhythmia classification model, the model with noise and the denoising model were compared, as listed in Table 5. The prediction F_1_-score of the original data for the ECG beat type was higher (average F_1_-score: 99.10%) than that of the noisy data. However, the prediction F_1_-score of the noisy data for the ECG beat type was lower than that of the original data (average F_1_-score: 25.04%). The predicted F_1_-score of the denoised data was higher than that of the noisy data and lower than, but close to, that of the original data (average F_1_-score: 95.71%). More information is provided in Appendix A.

The classification of the ECG signal mixed with noise showed that the F_1_-score exhibited an accuracy of 85% or higher in the case of the normal beat, which exhibited a high proportion compared with the other types of beats. Moreover, the classification results with denoised data showed a lower accuracy than the results using the original data. Nevertheless, arrhythmia could be classified with high accuracy, even with denoised data with an SNR of 30 or more.

### 4.3. Denoising and Classification of Measured ECG Data

Noise removal and arrhythmia classification were conducted using real measurement data. We compared the effectiveness of the proposed noise removal model by assessing arrhythmia classification outcomes, with and without noise removal. The classification was executed by medical professionals. The results indicated that the application of the proposed noise removal model yielded an average precision of 92.80%, an average recall of 90.46%, and an average F1-score of 91.60%. Conversely, without noise removal, the average precision was 67.89%, the average recall was 84.74%, and the average F1-score was 74.46% (see Table 6). These findings demonstrate that the proposed model significantly enhances arrhythmia classification performance. Further details are provided in Appendix A.

## 5. Discussion

In this study, data were continuously measured over four days in a practical setting through a wireless wearable electrocardiograph. The practical setting involved participants engaging in their daily routines, including working, exercising, and sleeping. The wearable device used, the Mezoo HiCardi SmartPatch, ensured that data collection was unobtrusive and reflective of the participants’ normal physiological conditions. This method enabled the collection of real-world data that accurately represent the variability and conditions encountered in everyday life. To the best of our knowledge, this is the first attempt at such extended-duration measurement using this technology. The data included irregular variations and were not representative of a controlled environment, which enabled us to capture the sudden occurrence of events over extended periods. Conventional R-peak-based segmentation involves several steps: identifying the R-peak, determining the cleavage point, and segmenting. In contrast, the proposed model segments a fixed time window simultaneously. Therefore, the procedure is simpler, involves real-time processing, and is suitable for feeding into a deep learning model. The proposed GAN-based noise removal model removed 0 dB noise with a magnitude exceeding 30 SNR. This can be attributed to the addition of connections to the residual blocks of U-Net, as well as distance information, to the generator loss function. Most previous studies [25,26] regarding noise removal from ECGs with a signal-to-noise ratio of 30 or more were limited to noise removal only, and no known studies on arrhythmia classification using noise-free ECG exist. The arrhythmia classification model, employing ResNet, outperformed previous models in regards to classification accuracy [27,28] and exhibited superior performance using ECG signals from which noise had been removed. The proposed model is modular, involving several processes, including fixed time-based segmentation, denoising, and arrhythmia classification. Hence, its performance could be maximized through the evaluation of each module. Furthermore, the developed noise removal and arrhythmia classification models demonstrated high performance using actual measured ECG signals. This serves as proof of the effectiveness of noise removal in arrhythmia classification.

This study can be compared with some previous studies [29,30,31] that similarly utilized wearable electrocardiographs, data collection for servers, application of noise filters, use of a convolutional neural network, learning with the public MIT-BIH databases, and their integration into an app service. Compared to these studies, this research incorporated methodological improvements. A neural network-based GAN filter, suitable for removing artifacts from ECG signals, was applied instead of a general-purpose band-pass filter or moving average filter, and transfer learning was employed using part of the directly measured data to enhance the discrimination of the actual data. This study also exhibits a comparative advantage in achieving state-of-the-art results in terms of noise removal and arrhythmia classification, as well as when these two functions are applied together. This suggests that wireless wearable ECG devices can be used for accurately detecting arrhythmia in noisy practical environments.

In summary, this study differs from previous studies in four aspects: (1) the utilization of labeled ECG data from wearable devices, (2) the application of filters specialized for removing ECG artifacts, (3) the classification of arrhythmia subtypes using a deep classifier and transfer learning, and (4) model modularization and integrated operation. Despite the various strengths of this study, there are limitations that could not be avoided. Unlike the R-peak-based segmentation, the 30 Hz (0.083 s) fixed time-based segmentation applied in this study may possess either none or more than two sets of label information for detecting arrhythmia within some sections. Because the training dataset was configured through augmentation with window rolling, it was eventually sufficient for learning aimed at classification. Comparing the results of this study, we found that the classifications from the public data and those from the measured data do not correspond exactly. Nevertheless, the model exhibited a high accuracy for the secured classes of the measured data.

We may attempt to extend the coverage of the proposed model. Although not necessarily for arrhythmia classification, it can also be considered for a separate utilization of the denoising model among modularized models. For example, it can be used before ST segmentation or monitoring, which also requires ECG plotting. The proposed research model is expected to aid follow-up studies.

Unlike most data used in modern deep-learning techniques, which comprise 2D images, the data in the health and medical fields are mostly one-dimensional. As a method for utilizing the data collected for healthcare and telemedicine, the proposed model can provide an appropriate utilization framework and can be actively applied not only for ECG but also in photoplethysmography.

## 6. Conclusions

We developed a model that can classify the subtypes of arrhythmia by removing noise in ECG signals collected by wearable devices in a practical setting. The model can be made available for real-time service. This model-based service can immediately provide highly accurate signal-related anomaly information to the individual wearing a wireless electrocardiograph, which can help the individual take evasive action or consider a follow-up.

## Figures and Tables

**Figure 1 sensors-24-05222-f001:**
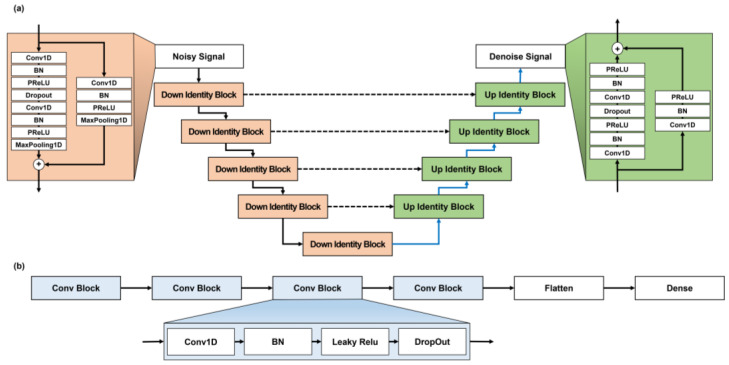
Model architecture used to denoise ECG signals: (**a**) generator model converting a noisy signal into a denoised signal. Layers of the same depth are connected in a U-Net structure composed of residual blocks. Each residual block convolution or deconvolution comprises a structure that builds up several layers and reconnects their inputs and outputs. (**b**) Discriminator model for generator learning, which helps regenerate the generator output. Abbreviations are as follows: Conv1D, 1D convolution layer; BN, batch normalization; Conv Transpose, convolution transpose; PReLU, parametric rectified linear unit; Leaky ReLU, leaky rectified linear unit.

**Figure 2 sensors-24-05222-f002:**
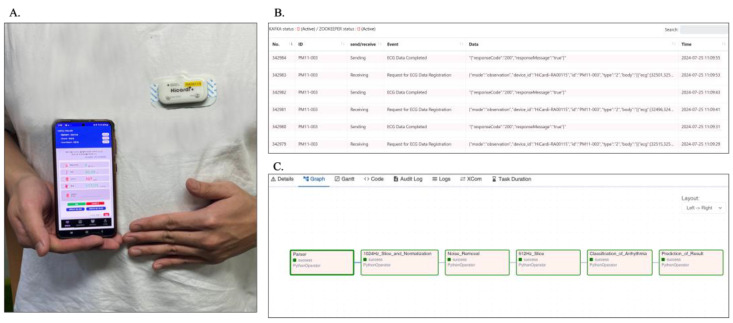
Detailed steps of the real-time ECG monitoring service implementation. (**A**): Smartphone app interface displays real-time ECG data received via Bluetooth from the Mezoo HiCardi SmartPatch wearable device. (**B**): Web server interface shows the streaming data events received from the smartphone app, illustrating the secure data transmission to the broker server via HTTPS. (**C**): Airflow interface depicts the data processing pipeline on the GPU server, including parsing, slicing, normalizing, noise removal, arrhythmia classification, and prediction of results.

**Figure 3 sensors-24-05222-f003:**
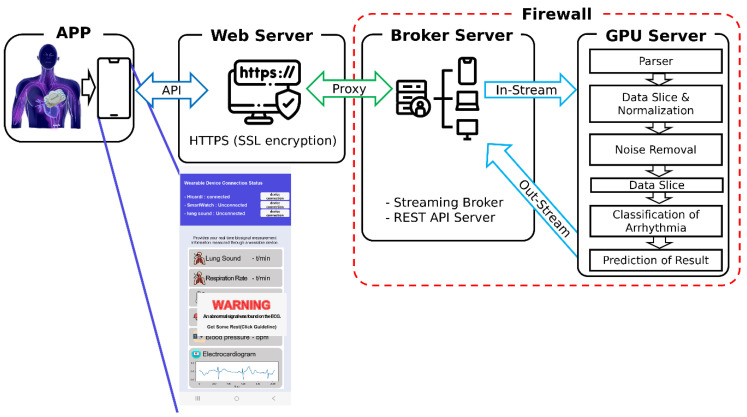
Real-time service scheme for arrhythmia classification through wireless wearable electrocardiograph. ECG is delivered by the broker to the security server through the smartphone of the service user, which is classified by artificial intelligence, and the classification result is again delivered to the smartphone of the user through the broker. Abbreviations used: HTTPS, hypertext transfer protocol secure; SSL, secure sockets layer; and API, application programming interface.

**Figure 4 sensors-24-05222-f004:**
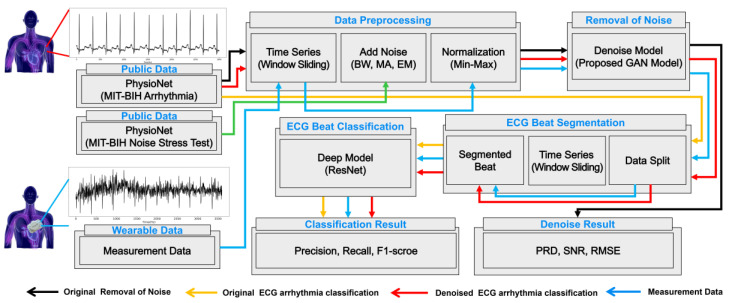
Schematic of data flow for classifying arrhythmia using ECG signals. Black lines correspond to the learning process of the denoising model using public data. Orange lines represent the training process of the arrhythmia classification model using the original ECG signals. Red lines indicate the process wherein noise is removed using the previously trained denoising model, followed by transfer learning on the pretrained arrhythmia classification model, and the classification results are displayed. Blue lines show the process of applying the same noise removal and transfer learning steps to data measured from wearable devices to derive classification results. Abbreviations are as follows: BW, baseline wander; MA, muscle artifacts; EM, electrode motion; GAN, generative adversarial network; PRD, percent root-mean-squared difference; SNR, signal-to-noise ratio; and RMSE, root-mean-squared error.

**Figure 5 sensors-24-05222-f005:**
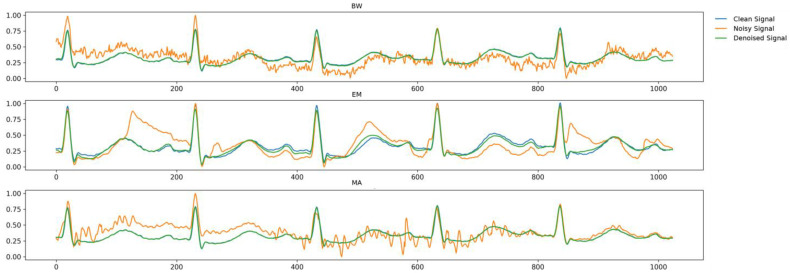
Noise removal results of the proposed model for each type of noise. Blue, orange, and green colors correspond to the clean, noisy, and denoised signals, respectively. Noise contains 0-dB standard. Abbreviations are as follows: BW, baseline wander; MA, muscle artifacts; EM, electrode motion.

**Figure 6 sensors-24-05222-f006:**
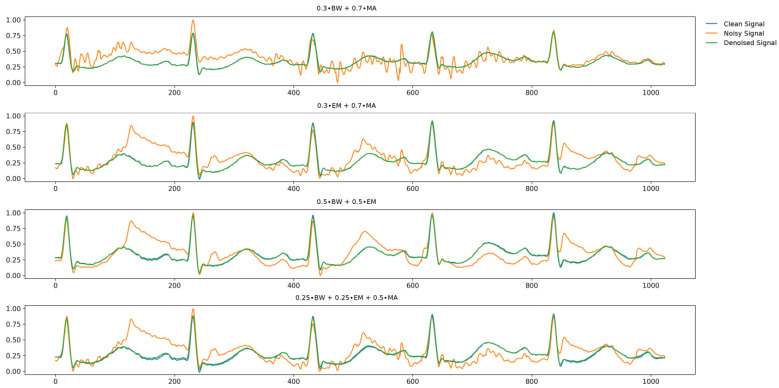
Noise removal results of the proposed model for each type of mixed noise. Blue, orange, and green colors correspond to the clean, noisy, and denoised signals, respectively. Noise contains 0-dB standard. Abbreviations are as follows: BW, baseline wander; MA, muscle artifacts; EM, electrode motion.

**Table 1 sensors-24-05222-t001:** Demographic characteristics of MIT-BIH and measurement data.

Variables	MIT-BIH (N = 47)	Measurement Data (N = 35)
Age (years)	67 (23–89)	68 (64–72)
Sex, N (%)		
Male	25 (53.2)	28 (80.0)
Female	22 (46.8)	7 (20.0)
Average measurement time	30 min	3 days, 16 h, 5 min

For continuous variables without a specific notation, the statistics of the median (Q1–Q3) are indicated. Q1, the first quartile; Q3, the third quartile.

**Table 2 sensors-24-05222-t002:** Beat frequencies from public ECG data.

Type of ECG Beat	Original Label	Proposed Label
N	74,790 (71.8)	235,851 (82.9)
B	15,334(14.7)	26,203 (9.2)
V	7124 (6.9)	14,466 (5.1)
A	2546 (2.4)	6264 (2.2)
F	803 (0.8)	1870 (0.7)
Total count	104,217 (100.0)	284,654 (100.0)

The values in parentheses indicate the column percentages. N: normal beat; B: interventricular block; V: premature ventricular contraction; A: atrial premature beat; F: ventricular fusion beats and normal beats.

**Table 3 sensors-24-05222-t003:** Comparison of ECG noise removal evaluation of each model for public ECG data.

		0 dB			5 dB		
		PRD (%)	SNR	RMSE	PRD (%)	SNR	RMSE
BW	Comparative model (U-Net generator)	14.45	16.82	0.066	12.59	18.00	0.058
	Proposed model (U-Net + LSGAN loss)	10.96	19.21	0.059	7.71	22.26	0.041
	Proposed model (U-Residual generator + LSGAN loss)	2.51	32.02	0.008	1.92	34.34	0.006
EM	Comparative model (U-Net generator)	16.48	15.67	0.076	13.68	17.28	0.063
	Proposed model (U-Net generator + LSGAN loss)	9.744	20.23	0.052	6.68	23.50	0.036
	Proposed model (U-Residual generator + LSGAN loss)	2.47	32.14	0.008	1.83	34.74	0.006
MA	Comparative model (U-Net generator)	14.76	16.62	0.068	11.61	18.71	0.053
	Proposed model (U-Net generator + LSGAN loss)	10.65	19.46	0.057	7.79	22.17	0.042
	Proposed model (U-Residual generator + LSGAN loss)	2.60	31.70	0.008	1.89	34.46	0.006

BW: baseline wander; EM: electrode motion; MA: muscle artifact; PRD: percent root-mean-squared difference; SNR: signal-to-noise ratio; RMSE: root-mean-squared error.

**Table 4 sensors-24-05222-t004:** Comparison of ECG mixed noise removal evaluation of each model for public ECG data.

	0 dB		
	PRD (%)	SNR	RMSE
0.3∙BW + 0.7∙MA	2.68	31.51	0.013
0.3∙EM + 0.7∙MA	2.70	31.38	0.012
0.5∙BW + 0.5∙EM	1.65	35.65	0.007
0.25∙BW + 0.25∙EM + 0.5∙MA	2.66	31.53	0.012

BW: baseline wander; EM: electrode motion; MA: muscle artifact; PRD: percent root-mean-squared difference; SNR: signal-to-noise ratio; RMSE: root-mean-squared error.

**Table 5 sensors-24-05222-t005:** Comparison of ECG arrhythmia classification evaluations of each model for public ECG data.

Type of ECG Beat	Data	Precision (%)	Recall (%)	F1-Score (%)
N	Noisy data	75.60	99.35	85.86
	Original data	99.87	99.98	99.92
	Denoised data	99.61	99.49	99.55
B	Noisy data	82.19	12.20	21.23
	Original data	99.99	100	99.97
	Denoised data	99.94	99.86	99.90
V	Noisy data	40.00	3.28	6.07
	Original data	99.63	98.36	98.99
	Denoised data	96.03	99.01	96.45
A	Noisy data	32.20	7.42	12.06
	Original data	99.61	98.83	99.22
	Denoised data	96.50	96.79	96.65
F	Noisy data	0.00	0.00	0.00
	Original data	98.25	96.55	97.39
	Denoised data	90.54	81.90	86.00
Average	Noisy data	46.00	24.45	25.04
	Original data	99.47	98.74	99.10
	Denoised data	96.52	95.41	95.71

N: normal beat; B: interventricular block; V: premature ventricular contraction; A: atrial premature beat; F: ventricular fusion beats and normal beats.

**Table 6 sensors-24-05222-t006:** Comparison of beat frequencies of measured ECG data and ECG arrhythmia classification evaluation of each model for measured ECG data.

Type of ECG Beat	Measurement Data	Data	Precision (%)	Recall (%)	F1-Score (%)
N	6,186,167 (97.82)	Original data	97.55	88.73	92.93
		Denoised data	97.55	91.84	94.61
B	23,309 (0.36)	Original data	66.39	85.24	74.65
		Denoised data	91.24	88.24	89.71
V	33,741 (0.53)	Original data	54.14	82.42	65.35
		Denoised data	93.15	92.21	92.68
A	80,723 (1.27)	Original data	53.47	82.56	64.90
		Denoised data	89.24	89.56	89.39
Average	855,811 (100.00)	Noisy data	67.89	84.74	74.46
		Denoised data	92.80	90.46	91.60

N: normal beat; B: interventricular block; V: premature ventricular contraction; A: atrial premature beat.

## Data Availability

The original contributions presented in the study are included in the article/Appendix A, further inquiries can be directed to the corresponding author.

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
