# Peer review of "Development and Validation of a Real-Time Service Model for Noise Removal and Arrhythmia Classification Using Electrocardiogram Signals"

_sensors, 2024, doi:10.3390/s24165222_

Round 1

Reviewer 1 Report

Comments and Suggestions for Authors

Manuscript ID: sensors-3053071

Title: Development and Validation of a Real-Time Service Model for Noise Removal and Arrhythmia Classification Using Electro-cardiogram Signals

This manuscript reported a deep learning real-time service model that utilizes Generative Adversarial Networks (GANs) for effective noise removal and ResNet for precise arrhythmia classification from wearable ECG data. Overall, the paper is well written and contains interesting and revealing content with clear proof; also, they utilized labeled ECG data, filters for removing ECG artifacts, used deep classifier & transfer learning for subtype classification to make a service model for integrated operation. I do not have any comments and I would like to recommend this manuscript for acceptance and further processing for publication.

Author Response

Thank you for your positive feedback and recommendation for the acceptance of our manuscript. We greatly appreciate your acknowledgment of our work and your supportive words.

Reviewer 2 Report

Comments and Suggestions for Authors

Referee letter:

In response to the fact that most of the studies on denoising of ECG signals have been limited to denoising, in this manuscript, Yeonjae Park et al. proposed a study on arrhythmia classification based on noise-free ECG signals. This study utilizes GANs for effective noise removal. The arrhythmia classification model employing ResNet exhibits high performance in classification accuracy.

This work developed a model that can classify subtypes of arrhythmia by removing noise in ECG signals collected by wearable devices in a practical setting. This model is a catalyst for the application of wearable devices for health monitoring.

In order to improve the manuscript for a re-submission or a submission to other journals, we have some suggestions for the authors as follows.

The main criticisms are:

Comment 1. In the introduction, authors find that there are already numerous articles utilizing AI solely for the purposes of ECG noise reduction or arrhythmia classification. Therefore, beyond integrating these two tasks, what additional compelling advantages does this article offer?

Comment 2. According to 3.1.2. Data Preprocessing, the more it is possible to encapsulate the variety of situations where noise interferes with the data in a real situation, the more favorable it should be for the actual measurement phase of the elimination will be. How the operation of adding noise is determined to be reasonable? Please explain more about how you think about determining the length of time the noise is added, the percentage of different types in the mix, and the selection of the level of noise interference.

Comment 3. In Figure 3, authors illustrate that the denoising model and the classification model are correlated. More details about how the two models are combined are suggested to be added, e.g. curious to know if denoised data are used in the training of the classification model, and if attempting to adjust the structure and parameters of the denoising model in turn based on the results of the classification?

Comment 4. In 2.4. Service Utilization of Developed Model, authors show how the real-time service work. In actual applications, how often can the real-time service give feedback at intervals? The article describes less about the implementation and results of the real-time service, so it is recommended to add some screenshots of each step in the application.

Comment 5. In table 5, please illustrate why is there a bigger gap between the classification results of fusion of the ventricles and normal beats based on original data and denoised data as compared to the other ones?

Comment 6. In abstract, authors state that ECG monitoring is susceptible to noise interference so noise removal is important. It is curious whether the model for denoising in this paper be generalized to the processing of other electrical signal acquisitions. And how do the authors think the artifact problem will be solved in the future?

Comment 7. The explanation of the abbreviation located in Line 243 is incorrect, please check the abbreviation in the whole text for the same error.

Comment 8. The specific images of the ECG are placed in the supporting material causing the manuscript to be a bit monotonous, please consider placing some of the important images in the body of the text.

Comments on the Quality of English Language

None

Author Response

Comment 1: In the introduction, authors find that there are already numerous articles utilizing AI solely for the purposes of ECG noise reduction or arrhythmia classification. Therefore, beyond integrating these two tasks, what additional compelling advantages does this article offer?

Response 1: We sincerely thank Reviewer #2 for their valuable comments. In response to the reviewer’s inquiry regarding the compelling advantages this study offers beyond integrating ECG noise reduction and arrhythmia classification, we cite several notable strengths of this research as follows:

  1. Utilization of directly measured datasets: Unlike previous studies that primarily used public datasets for model performance comparison, this study directly employs the measured datasets for external validation. Therefore, the proposed approach enhances the generalizability and applicability of our results in real-world settings.
  2. Comprehensive evaluation and additional validation: The noise reduction performance was verified not only through standard metrics such as PRD, SNR, and RMSE but also through the comparison and analysis of noisy signals, original signals, and denoised signals in the context of arrhythmia classification. This comprehensive evaluation method demonstrates the practical impact of noise reduction on the accuracy of arrhythmia detection.

These aspects contribute to the reliability and scalability of our research findings, which distinguishes this study from the existing literature. We added the following content to the manuscript:

  • In contrast to previous research, this study directly used the measured data for ex-ternal validation. The noise removal performance was verified through simple evaluation metrics, as well as via comparative analysis of the signals with noise, original signals, and noise-removed signals for arrhythmia classification. (Line 84 – 88)

Comment 2: According to 3.1.2. Data Preprocessing, the more it is possible to encapsulate the variety of situations where noise interferes with the data in a real situation, the more favorable it should be for the actual measurement phase of the elimination will be. How the operation of adding noise is determined to be reasonable? Please explain more about how you think about determining the length of time the noise is added, the percentage of different types in the mix, and the selection of the level of noise interference.

Response 2: We are particularly thankful to Reviewer #2 for their insightful comments. We appreciate the opportunity to elaborate on our method of adding noise and the rationale behind it, as discussed in Section 3.1.2 on Data Preprocessing. Below, we provide a detailed explanation of the noise addition procedure and its justification:

  1. Justification for Adding Noise: This study aims to remove noise from electrocardiogram (ECG) measurements and accurately classify arrhythmias based on the denoised ECG. The PhysioNet data is typically provided with the noise already removed and basic preprocessing completed. However, to ensure that our model can robustly handle various types of noise that may occur in real-world settings, it is essential to train the model using data with artificially added noise. Accordingly, we artificially added noise to the data, and treated the noisy data as X and the original noise-free data as Y. This method of training is consistent with approaches used in numerous existing studies.
  2. Determination of Noise Addition Duration: The duration of noise addition was determined with reference to the MIT-BIH Arrhythmia Database and the MIT-BIH Noise Stress Test Database. The noise types labeled as BW, MA, and EM in the Noise Stress Test Database each have a duration of 30 min, which matches the length of the data in the Arrhythmia Database. This consistency in duration enables the effective synthesis of the noises with the original data. To clarify this in the manuscript, we included the following statement:
  • For denoising, the original data and each noise segment are shifted to 180 Hz (0.50 s) and then cut at 1024 Hz (2.84 s) to synthesize the noise. Additionally, the segmented original data is divided into two intervals, 0-512 Hz and 513-1024 Hz, each at 512 Hz (1.42 s), for arrhythmia classification. (Line 258 – 261)
  1. Selection of the Level of Noise Interference: The ratios of mixed noise types were set based on the likelihood of encountering various noises in real-world environments. We utilized the following ratios: 0.3·BW + 0.7·MA, 0.3·EM + 0.7·MA, 0.5·BW + 0.5·EM, and 0.25·BW + 0.25·EM + 0.5·MA. These ratios were selected because MA noise was considered the most prevalent type. Consequently, MA noise was assigned higher weights to more accurately simulate its impact in realistic scenarios. As stated in the manuscript, " During the synthesis process, we placed higher importance on the MA noise by as-signing it a higher weight, as we believed it to be the most significant type of noise.”(Line 308 – 310)
  2. Selection of the Level of Noise Interference: The levels of noise interference were selected based on findings from previous studies, which tested various noise intensity levels such as 0 dB, 1.25 dB, 3 dB, and 5 dB. Most studies predominantly used 0 dB and 5 dB. Accordingly, we implemented noise interference levels of 0 dB and 5 dB in our study to ensure the robust performance of the model under diverse noise conditions.

Comment 3: In Figure 3, authors illustrate that the denoising model and the classification model are correlated. More details about how the two models are combined are suggested to be added, e.g. curious to know if denoised data are used in the training of the classification model, and if attempting to adjust the structure and parameters of the denoising model in turn based on the results of the classification?

Response 3: We sincerely thank Reviewer #2 for their insightful comments. We deeply appreciate your feedback concerning the detailed explanation of how the two models are integrated. As you noted, the manuscript initially lacked sufficient detail in this aspect. In response, we have revised the manuscript to include a more comprehensive description of how the denoising model and the classification model are combined. Additionally, due to the addition of a new figure in the manuscript, the original Figure 3 has been renumbered to Figure 4.

  • This study entailed several integrated steps: data sourcing and preprocessing, noise removal, bit segmentation, arrhythmia classification, and performance evaluation for both noise removal and ECG arrhythmia classification. Initially, the original ECG signals, sourced from public databases (MIT-BIH Arrhythmia and MIT-BIH Noise Stress Test from PhysioNet) and wearable devices, undergo preprocessing. During this phase, a time-series data window sliding technique is utilized, signals are normalized using the Min–Max method, and the noise removal model is trained and evaluated using specific indices. Subsequently, for the arrhythmia classification model, the pre-processed signals are segmented into bits using the time-series data window sliding technique. These bits are then trained using a deep learning model based on ResNet to classify arrhythmias, with verification based on evaluation indices. To assess the difference in arrhythmia classification performance between the original and noise-removed signals, a proposed GAN model is employed to eliminate noise. Thereafter, the noise-removed signal is processed through the same ECG beat seg-mentation step and trained using the existing arrhythmia classification model, followed by model verification based on evaluation indices. For practical application, noise is removed from directly measured data using the same method, and the existing arrhythmia classification model is adapted to finalize the model. The completed model is evaluated based on various performance indicators to assess its real-world applicability. This process is depicted in Fig. 4, which visually outlines the flow and method-ology of each step.(Line 202 – 221)

Additionally, we have enhanced the clarity of Figure 4 as follows:

Figure 4. Schematic of data flow for classifying arrhythmia using ECG signals. Black lines correspond to the learning process of the denoise model using public data. Orange lines represent the training process of the arrhythmia classification model using the original ECG signals. Red lines indicate the process wherein noise is removed using the already trained denoise model, followed by transfer learning on the pretrained arrhythmia classification model, and the classification results are displayed. Blue lines show the process of applying the same noise removal and transfer learning steps to data measured from wearable devices to derive classification results. Abbreviations are as follows: BW, baseline wander; MA, muscle artifacts; EM, electrode motion; GAN, generative adversarial network; PRD, percent root-mean-squared difference; SNR, signal-to-noise ratio; and RMSE, root-mean-squared error.

Comment 4: In 2.4. Service Utilization of Developed Model, authors show how the real-time service work. In actual applications, how often can the real-time service give feedback at intervals? The article describes less about the implementation and results of the real-time service, so it is recommended to add some screenshots of each step in the application.

Response 4: We extend our gratitude to Reviewer #2 for their insightful comments. We have incorporated additional explanations in the main text addressing the points raised. Furthermore, to facilitate understanding, we have included screenshots of each step in the application process in Figure 2. Additionally, due to the addition of a new figure in the manuscript, the original Figure 2 has been renumbered to Figure 3.

  • The wearable device (Mezoo HiCardi SmartPatch) collects real-time ECG data from the body, which is transmitted via Bluetooth to a smartphone app. This app serves as a user interface and forwards the data to the web server via API communication (Fig. 2-A). The web server securely transmits the data to the broker server using the HTTPS protocol. The broker server, functioning as a streaming broker and REST API server (Fig. 2-B), initiates data analysis by streaming it to the GPU server. Here, the data undergoes parsing, slicing, normalization, noise removal, and arrhythmia classification to predict the final results (Fig. 2-C). The processed results are then streamed back through the broker server and displayed on the smartphone, enabling real-time ECG monitoring. The system employs Airflow (2.2.5) for data flow management, Kafka (2.12-2.8.1) as the broker server, and TensorFlow (2.4.1) as the deep learning framework. The overall system architecture is illustrated in Fig. 3.(Line 174 – 185)

Figure 2. Detailed steps of the real-time ECG monitoring service implementation. A: Smartphone app interface displays real-time ECG data received via Bluetooth from the Mezoo HiCardi SmartPatch wearable device. B: Web server interface shows the streaming data events received from the smartphone app, illustrating the secure data transmission to the broker server via HTTPS. C: Airflow interface depicts the data processing pipeline on the GPU server, including parsing, slicing, normalizing, noise removal, arrhythmia classification, and prediction of results.

Comment 5: In table 5, please illustrate why is there a bigger gap between the classification results of fusion of the ventricles and normal beats based on original data and denoised data as compared to the other ones?

Response 5: We sincerely thank Reviewer #2 for their insightful comments. We believe the notable performance discrepancy between the Fusion of the Ventricles and Normal Beats (F) label and other labels can be attributed to the following factors:

  1. Data Imbalance: The fusion of the ventricles and normal beats (F) constitutes only 0.6% of the entire dataset. This significant imbalance poses challenges for the learning and classification capabilities of the model, particularly in the presence of data noise.
  2. Complex Beat Characteristics: The fusion of the ventricles and normal beats label exhibits characteristics of both ventricles and normal beats. This complexity complicates the ability of the model to accurately distinguish this label from others.

These factors likely contribute to the observed performance gap between the fusion of the ventricles and normal beats label and other labels.

Comment 6. In abstract, authors state that ECG monitoring is susceptible to noise interference so noise removal is important. It is curious whether the model for denoising in this paper be generalized to the processing of other electrical signal acquisitions. And how do the authors think the artifact problem will be solved in the future?

Response 5: We sincerely thank Reviewer #2 for their valuable comments and acknowledge the importance of noise removal, as ECG monitoring is susceptible to noise interference. Although the proposed noise removal model is tailored for bioelectrical signals such as ECG, it has potential applicability to other types of electrical signal acquisition, especially when the fundamental characteristics of the electrical signal and the noise pattern are similar. However, additional verification and adjustments are necessary for each signal type. In response to the reviewer's query regarding addressing artifacts, we propose several approaches:

  1. The development of more sophisticated signal processing algorithms and machine learning techniques, which will enhance the distinction between noise and valid signals. Notably, recent studies employing the diffusion model, which follows the development of generative adversarial networks (GANs), are addressing issues inherent in existing GANs, suggesting technological progress.
  2. A multimodal approach that integrates multiple signal sources could significantly aid in artifact removal.
  3. On the hardware front, advancements in signal acquisition equipment and technologies to minimize environmental noise are crucial. Despite the progress in algorithms, hardware improvements are fundamental, and various algorithms can enhance these hardware capabilities.

Combining these efforts is expected to enhance future noise removal models and extend their applicability to diverse electrical signal processing tasks.

Comment 7: The explanation of the abbreviation located in Line 243 is incorrect, please check the abbreviation in the whole text for the same error.

Response 7: We sincerely thank Reviewer #2 for their valuable comments. We have revised the manuscript to incorporate this feedback.

Comment 8: The specific images of the ECG are placed in the supporting material causing the manuscript to be a bit monotonous, please consider placing some of the important images in the body of the text.

Response 8: In response to your suggestion to enhance the visual appeal of the manuscript by integrating key ECG images within the main text, we have made the necessary adjustments. Several pivotal ECG images have now been embedded directly into the text to make the manuscript more engaging and informative. We believe this modification will provide better context and clarity for the readers.

Figure 5. Noise removal results of the proposed model for each type of noise. Blue, orange, and green colors correspond to the clean, noisy, and denoised signals, respectively. Noise contains 0-dB standard. Abbreviations are as follows: BW, Baseline wander; MA, muscle artifacts; EM, electrode motion;

Figure 6. Noise removal results of the proposed model for each type of mixed noise. Blue, orange, and green colors correspond to the clean, noisy, and denoised signals, respectively. Noise contains 0-dB standard. Abbreviations are as follows: BW, baseline wander; MA, muscle artifacts; EM, electrode motion;

We express our gratitude once again to Reviewer #2 for their insightful comments that significantly improved the quality of the manuscript.

Reviewer 3 Report

Comments and Suggestions for Authors

The paper describes an interesting framework based on a deep learning model for the analysis of ECG signals acquired from wearable devices. Wearable devices have opened new and promising avenues for the management of cardiovascular diseases. Simultaneously, the use of AI and deep learning algorithms is becoming more and more popular, enabling signal analysis and real-time processing that could only be imagined a few years ago. Even though this growing interest has led to the proliferation of many papers and research activities in this field, this paper presents some novel elements, well highlighted by the authors. Thus, I believe that it is of interest to the scientific community in general and to the readers of Biosensors in particular. The paper is well-written and well-referenced. There are only a few points I think should be addressed:

1.           My major concern is about the description of how the model has been tested on ECG signals acquired from the wearable patch (Mezoo HiCardi SmartPatch). First of all, I believe that additional details about this device could be useful for the reader (e.g., is it a medical device? Is it a single-lead ECG? How long does each ECG recording last? Does it implement any ECG processing before sending the data to the server?). Additionally, it is not very clear how the performance of the ECG arrhythmia classification obtained from the acquired ECG has improved thanks to the use of the deep learning models. In Table 6, it seems that only the performance of the final classification—after implementing the deep learning models—is shown. It is not immediately clear how these results compare with those that would be obtained without the denoising.

2.           A second point, if I understand correctly, is that the development of the model was achieved by training and then testing it. No validation was performed. To my knowledge, particularly for deep learning models that utilize GANs, the validation phase is important in determining the best performance of the system. Why did the authors decide not to perform the validation of their model?

Other minor points:

-             Line 81: Even though RasNet is now a well-known and established technique for classification models, adding a little more detail on how it works could be useful for the general reader.

-             Line 100: Why is it necessary to resample the signal to match the sampling frequency of the data in the MIT-BIH database? Isn’t the model able to work with signals at different sampling frequencies?

-             Line 151: Is the “wearable device” mentioned the Mezoo HiCardi SmartPatch previously mentioned? Using the same name can help avoid confusion.

-             Line 290: More details about the “practical setting” where the data were collected should be provided.

Author Response

Comment 1: My major concern is about the description of how the model has been tested on ECG signals acquired from the wearable patch (Mezoo HiCardi SmartPatch). First of all, I believe that additional details about this device could be useful for the reader (e.g., is it a medical device? Is it a single-lead ECG? How long does each ECG recording last? Does it implement any ECG processing before sending the data to the server?). Additionally, it is not very clear how the performance of the ECG arrhythmia classification obtained from the acquired ECG has improved thanks to the use of the deep learning models. In Table 6, it seems that only the performance of the final classification—after implementing the deep learning models is shown. It is not immediately clear how these results compare with those that would be obtained without the denoising.

Response 1: We sincerely thank Reviewer #3 for their valuable comments. We  appreciate your suggestion to provide additional details about the wearable patch (Mezoo HiCardi SmartPatch) used in our study. We have included the following specific information in the manuscript:

  • We utilized the Hicardi® (MEZOO Co., Ltd., Wonju-si, Gangwon-do, Republic of Korea), a wearable ECG monitoring patch weighing 8 grams and measuring 42 mm × 30 mm × 7 mm. This device is certified by the Korea Food and Drug Safety Agency and captures ECG signals at a sampling frequency of 250 Hz with a resolution of 14 bits. It can monitor and record single-lead ECG, respiration, skin surface temperature, and physical activity. Data from the wearable patch were transmitted to a mobile gateway via Bluetooth Low Energy, implemented as a portable smartphone application. All collected data were then forwarded to a cloud-based monitoring server through the mobile gateway. After obtaining informed consent from all participants, the wearable patch was affixed to the left sternal border, and continuous recording of ECG signals and other data ensued. These data were subsequently reviewed by cardiologists using the cloud-based monitoring system. (Line 101 – 112)

Additionally, we concur with the reviewer’s observation regarding the difficulty in discerning the improvement in ECG arrhythmia classification performance by the deep learning model from Table 6 alone. In agreement with this point, we have revised Table 6 in the main text to include results without noise removal for comparative analysis, as follows:

Table 6. Comparison of beat frequencies of measured ECG data and ECG arrhythmia classification evaluation of each model on measured ECG data.

Type of ECG beat

Measurement data

Data

Precision (%)

Recall (%)

F1-score (%)

N

6,186,167 (97.82)

Original data

97.55

88.73

92.93

Denoised data

97.55

91.84

94.61

B

23,309 (0.36)

Original data

66.39

85.24

74.65

Denoised data

91.24

88.24

89.71

V

33,741 (0.53)

Original data

54.14

82.42

65.35

Denoised data

93.15

92.21

92.68

A

80,723 (1.27)

Original data

53.47

82.56

64.90

Denoised data

89.24

89.56

89.39

Average

855,811 (100.00)

Original data

67.89

84.74

74.46

Denoised data

92.80

90.46

91.60

N: normal beat, B: interventricular block, V: premature ventricular contraction, A: atrial premature beat.

We have revised the results section of the manuscript as follows:

  • Noise removal and arrhythmia classification were conducted using real measurement data. We compared the effectiveness of the proposed noise removal model by assessing arrhythmia classification outcomes with and without noise removal. The classification was executed by medical professionals. The results indicated that the application of the proposed noise removal model yielded an average precision of 92.80%, an average recall of 90.46%, and an average F1-Score of 91.60%. Conversely, without noise removal, the average precision was 67.89%, the average recall was 84.74%, and the average F1-Score was 74.46% (refer to Table 6). These findings demonstrate that the proposed model significantly enhances arrhythmia classification performance. Further details are provided in Figure S11. (Line 347 – 356)

Comment 2: A second point, if I understand correctly, is that the development of the model was achieved by training and then testing it. No validation was performed. To my knowledge, particularly for deep learning models that utilize GANs, the validation phase is important in determining the best performance of the system. Why did the authors decide not to perform the validation of their model?

Response 2: We extend our gratitude to Reviewer #3 for their constructive feedback. Typically, in the development of machine learning and deep learning models, data is partitioned into training, validation, and testing sets for model training and evaluation. In the cited paper, the model was trained and evaluated using only the training and testing sets. Similarly, the TensorFlow GAN tutorial we followed did not incorporate a validation phase. Consequently, we adopted this approach. Nevertheless, we acknowledge your insightful suggestions and will consider including a validation phase in future research to ensure a more comprehensive evaluation in future.

Comment 3: Line 81: Even though ResNet is now a well-known and established technique for classification models, adding a little more detail on how it works could be useful for the general reader.

Response 3: We also thank Reviewer #3 for the suggestion to elaborate on the workings of ResNet for the general reader. We have updated the manuscript to include a detailed explanation of ResNet. Instead of adding this information to the Introduction, we have integrated it into the section on Arrhythmia Classification using ECG Signals, where the application of ResNet is discussed.

  • The residual block of ResNet comprises several essential components. First, each residual block includes an identity shortcut that bypasses one or more convolutional layers. This shortcut ensures that information is preserved as it passes through the layers by directly adding the input of the block to its output, which helps mitigate the vanishing gradient problem during training. Second, each residual block typically contains two convolutional layers that are crucial for feature extraction and transformation. The first convolutional layer processes the input data, followed by batch normalization and a ReLU activation function. Subsequently, the second convolutional layer processes the data further, followed by another batch normalization and ReLU activation. The output of this second layer is then added to the original input to produce the block's output. This architecture aids in alleviating the vanishing gradient problem that can occur as the network deepens, thus enabling the effective training of deeper networks. (Line 160 – 172)

Comment 4. Line 100: Why is it necessary to resample the signal to match the sampling frequency of the data in the MIT-BIH database? Isn’t the model able to work with signals at different sampling frequencies?

Response 4: We sincerely thank Reviewer #3 for their valuable comments. Regarding the necessity of resampling the signal to match the sampling frequency of the data in the MIT-BIH database, adjusting the length to 1024 without resampling would mean that data sampled at 250 Hz would span 4.096 s, whereas the data sampled at 360 Hz would span 2.84 s. Consequently, this would lead to the model processing data of varying time lengths. To ensure consistent input shape and duration, the signal needs to be resampled to a uniform sampling frequency. This uniformity is vital for optimizing performance. Therefore, resampling the signal to match the sampling frequency of the data in the MIT-BIH database is essential. We have revised the manuscript to include this explanation.

  • The data were initially measured at 250 Hz but were up-sampled to 360 Hz using the Fourier method to match the frequency of the data in the MIT-BIH database. This adjustment ensures the temporal consistency of the input data and secures a uniform input shape. (Line 112 – 114)

Comment 5:  Line 151: Is the “wearable device” mentioned the Mezoo HiCardi SmartPatch previously mentioned? Using the same name can help avoid confusion.

Response 5: We sincerely thank Reviewer #3 for their valuable comments. We have revised the manuscript to include this information regarding the wearable device (Mezoo HiCardi SmartPatch).

  • Wearable device (Mezoo HiCardi SmartPatch) (Line 202)

Comment 6.  Line 290: More details about the “practical setting” where the data were collected should be provided.

Response 6: We sincerely thank Reviewer #3 for their valuable comments. we have revised the manuscript to include this information.

  • In this study, data were continuously measured over four days in a practical setting through a wireless wearable electrocardiograph. The practical setting involved participants engaging in their daily routines, including working, exercising, and sleeping. The wearable device used, the Mezoo HiCardi SmartPatch, ensured that data collection was unobtrusive and reflective of the participants' normal physiological conditions. This method enabled the collection of real-world data that accurately represents the variability and conditions encountered in everyday life. (Line 363 – 369)

Finally, we would like to extend our gratitude to Reviewer #3 for their excellent comments, which significantly enhanced the quality of the manuscript.

Round 2

Reviewer 2 Report

Comments and Suggestions for Authors

After the revison, the manuscript can be accepted.

Comments on the Quality of English Language

None

Reviewer 3 Report

Comments and Suggestions for Authors

The authors have adequately addressed all my observations. I have no further comments or clarifications to make.